# *In vitro* genome editing activity of Cas9 in somatic cells after random and transposon-based genomic Cas9 integration

Jenny-Helena Söllner[1], Hendrik Johannes Sake[1], Antje Frenzel[1], Rita Lechler[1], Doris Herrmann[1], Walter Fuchs[2], Björn Petersen[1]*

1 Friedrich-Loeffler-Institut, Institute for Farm Animal Genetics, Neustadt am Rübenberge, Lower Saxony, Germany, 2 Friedrich-Loeffler-Institut, Institute of Molecular Virology and Cell Biology, Greifswald, Mecklenburg-Western Pomerania, Germany

* Bjoern.petersen@fli.de

**Data Availability Statement:** All additional data for NGS, qPCR, and Flow Cytometry have been made available under: https://osf.io/sxft2/.

## Abstract

Due to its close resemblance, the domesticated pig has proven to be a diverse animal model for biomedical research and genome editing tools have contributed to developing porcine models for several human diseases. By employing the CRISPR-Cas9 system, porcine embryos or somatic cells can be genetically modified to generate the desired genotype. However, somatic cell nuclear transfer (SCNT) of modified somatic cells and embryo manipulation are challenging, especially if the desired genotype is detrimental to the embryo. Direct *in vivo* edits may facilitate the production of genetically engineered pigs by integrating Cas9 into the porcine genome. Cas9 expressing cells were generated by either random integration or transposon-based integration of Cas9 and used as donor cells in SCNT. In total, 15 animals were generated that carried a transposon-based Cas9 integration and two pigs a randomly integrated Cas9. Cas9 expression was confirmed in muscle, tonsil, spleen, kidney, lymph nodes, oral mucosa, and liver in two boars. Overall, Cas9 expression was higher for transposon-based integration, except in tonsils and liver. To verify Cas9 activity, fibroblasts were subjected to *in vitro* genome editing. Isolated fibroblasts were transfected with guide RNAs (gRNA) targeting different genes (GGTA1, B4GALNT2, B2M) relevant to xenotransplantation. Next generation sequencing revealed that the editing efficiencies varied (2–60%) between the different target genes. These results show that the integrated Cas9 remained functional, and that Cas9 expressing pigs may be used to induce desired genomic modifications to model human diseases or further evaluate *in vivo* gene therapy approaches.

## Introduction

Over the years, the domesticated pig has shown to be invaluable as a protein source for human consumption and a diverse animal model for biomedical research. Pigs show a higher anatomical, physiological, and genetic resemblance to humans than rodents, which still represent the

**Funding:** JHS was funded within the framework of an intrainstitutional African Swine Fever research consortium.

**Competing interests:** The authors have declared that no competing interests exist.

main human disease model [1–3]. Due to these similarities, certain disease progressions are more adequately mimicked in porcine models [4]. Pig models have been developed for a huge variety of human diseases such as cystic fibrosis [5–7], diabetes [8–10], cancer [11–13], X-linked severe combined immunodeficiency [14, 15], and Duchenne muscular dystrophy [16]. While pork production and biomedical research may require different genotypes or phenotypes, recent genetic engineering developments provide benefits for both industries. For example, specific genetic modifications are critical for xenotransplantation to avoid undesired immune reactions towards the porcine donor organ [17]. Recently, the U.S. Food and Drug Administration (FDA) approved pigs (GalSafe) with intentional genomic alteration for meat consumption which potentially can be used for biomedical products such as xenografts for humans [18].

Genome editing technologies such as CRISPR-Cas9 revolutionized targeted genetic engineering in livestock species by their simplicity and efficiency. CRISPR (clustered regularly interspaced short palindromic repeats) and its associated endonuclease protein Cas9 induce double-stranded breaks (DSBs) in the DNA [19] and have been frequently used to modify the genome of animals over the years. Cas9 is guided by a guide RNA (gRNA) recognizing complementary DNA strands. After binding, the Cas9 cleaves the DNA at the target site. However, while *in vitro* modification became more efficient, the production of genetically modified (GM) pigs remains challenging. GM pigs are either produced by somatic cell nuclear transfer (SCNT) or via micromanipulation or electroporation of zygotes, all are laborious, and require a high expertise. Hence, new means to increase the production efficiency of GM animals are sought. *In vivo* modifications can be achieved when the Cas9 is integrated into the pig's genome and gRNAs are directly delivered into specific tissues of living animals. Thereby, germline modifications can be avoided to establish new disease models [20, 21], generate genotypes which are detrimental to embryos, or evaluate approaches for *in vivo* gene therapies.

In mice, several attempts of *in vivo* tissue modifications have already proven to be successful [22–24]. *In vivo* genome editing by injecting Cas9 and selected gRNAs into mice was applied to investigate for example Wolff-Parkinson-White syndrome [22] or lung cancer development [23, 24]. However, the delivery of Cas9 and gRNAs comes with challenges due to the size of the Cas9 expression cassette [25]. Hence, transgenic mice expressing Cas9 were developed to induce the desired genomic modification by only delivering the gRNAs *in vivo* [20]. To advance the development of pig models for human diseases, Cre-dependent Cas9 expressing pigs were first generated in 2017, showing successes of *in vivo* genome editing in pigs [21]. Several tumor-suppressing cells were targeted and inactivated by delivering specific gRNAs into the Cas9 expressing pigs, causing tumor formation in different organs [21]. Also, results of targeted Cas9 integration and its functionality in fibroblast, porcine adipose-derived mesenchymal stem cells, and porcine organoids were recently reported [26]. Maintaining breeding lines of Cas9 expressing animals provides opportunities to avoid SCNT or manipulation of early embryos. Therefore, we generated pigs carrying a Cas9 integration by two different approaches and assessed their functionality *in vitro*.

## Material and methods

### Ethics statement

All experiments involving animals were approved by the responsible authority ('Landesamt für Verbraucherschutz und Lebensmittelsicherheit' in Lower Saxony, Germany, Animal Experiment No.: TVA 33.8-42502-04-18/2862 and TVA 33.8-42502-04-16/2343). Animals were housed and cared for according to German Animal Welfare regulations.

## Generation of Cas9 expressing pigs

**Establishment of donor cells.**   Fetal male fibroblasts were cultured as previously described [27, 28]. In short, culture media consisted of DMEM (Dulbecco´s modified Eagle's medium) (Capricorn Scientific) supplemented with 2 mM L-glutamine (AppliChem), 0.1 mM mercaptoethanol, 1% 100x penicillin/streptomycin (Pen/Strep), 1% 100x non-essential amino acid, and 1% 100x sodium pyruvate (Sigma-Aldrich), and 10–30% Fetal bovine serum (FBS) (Capricorn-Scientific). The commonly used pX330-U6- Chimeric_BB-CBh-hSpCas9 (pX330) a gift from Feng Zhang (Addgene plasmid # 42230; http://n2t.net/addgene:42230; RRID:Addgene_42230) [29] was used for transfection of porcine fetal fibroblasts to establish a random integration (RI) of Cas9. The vector was previously modified to express a neomycin selection marker (Fig 1). For transposon integration of Cas9, a Sleeping Beauty (SB) transposon vector system was designed. The transgenes Cas9, the gRNA array, and neomycin were flanked by inverted repeats to bind the SB transposase. The SB transposase vector pCMV(CAT)T7-SB100 was a gift from Zsuzsanna Izsvak (Addgene plasmid # 34879; http://n2t.net/addgene:34879; RRID: Addgene_34879) [30]. The SB transposon construct (625 ng/µl) and the SB transposase (595 ng/µl) vector were co-transfected. Cultured cells ($3x10^6$) were transfected with a total 5 µg circular plasmid solution. Fibroblast were electroporated with the Neon Transfection System (Invitrogen, Thermo Fisher Scientific) with settings set to 2 x 20ms pulses at 1350 V. Following transfection, cells were cultured in antibiotic-free media for 24 hours. After 24 hours, cells were cultured and selected for neomycin resistance with G418 (800 µg/ml) (Carl Roth) for 10 days.

**Characterization of transfected fibroblasts.**   Selected fibroblasts were treated with lysis buffer (0.02% SDS, 20 mM Tris-HCL) containing proteinase K (50 µg/ml) (Thermo Fisher Scientific) to extract genomic DNA for polymerase chain reaction (PCR). Cas9 was amplified by the following primers: forward primer 5' `ACAAGCTGATCCGGGAAGTG` 3' and reverse primer 5' `ACAAGCTGATCCGGGAAGTG` 3'. The selection marker neomycin was amplified with 5' `CAGGATGATCTGGACGAAGA` 3' and 5' `GATGCGCTGCGAATCGGGAG` 3' and the gRNA cassette with 5' `ATGCTTACCGTAACTTGAAAG` 3' and 5' `ATTTGTCTGCA GAATTGGCG` 3' [31].

**Somatic cell nuclear transfer.**   Porcine ovaries were collected at a local slaughterhouse. Aspiration of follicles and maturation of oocytes have been previously described [32]. Briefly, oocytes were matured for 40 hours in maturation media containing FGF2 (Peprotech), LIF (ESGRO Mouse LIF), and IGF1 (R&D Systems) [33]. Cas9 expressing cells were used as donor cells for SCNT. The SCNT protocol has been described in detail by Hölker et. al., (2005) and Petersen et. al., (2008). In short, the metaphase II oocytes were subjected to enucleation by removing the polar body and metaphase plate, followed by fusion of the donor cell and oocyte by an electric pulse and subsequent culture for 20 hours until transfer [27, 28]. Eight German Landrace gilts (7–9 month) were hormonally synchronized with a standard superovulation protocol [28]. Fifty-nine to eighty-five one-two-cell-stage embryos were transferred into gilts. Recipients were checked for pregnancy on day 25 post op by ultrasound scanning. Six gilts established and maintained a pregnancy.

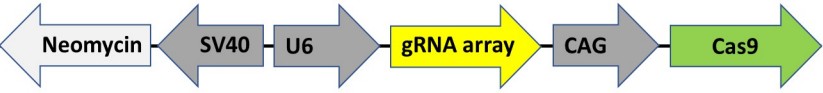

**Fig 1. Transgenes inserted into the porcine genome.** A neomycin selection marker driven by the S40 promoter was inserted into the vector to select for fibroblasts with desired vector integration. The gRNA array and Cas9 were driven by a U6 and CAG promoter, respectively.

## Genotyping offspring

DNA of the piglets was extracted from tail samples. About 50 mg of tail tissue was lysed in tail lysis buffer (50 mM Tris-HCL, 100 mM NaCl, 100 mM EDTA, 1% SDS, and 40 μl 10 mg/ml proteinase K) overnight at 50˚C, followed by ethanol precipitation. The samples were eluted in aqua bidest and diluted to a concentration of 20 ng/μl for PCR characterization. PCRs for Cas9, neomycin and gRNA integration were performed as described above. The Cas9 amplicon was purified for Sanger sequencing with Invisorb® Fragment CleanUp (Invitek Molecular GmbH) diluted to 20 ng/μl and 5 μM of primer was added.

**Reverse-transcription qPCR.**   To determine Cas9 expression in several organs, two boars were sacrificed, one boar with a SB transposon-based integration (SB pig) and one boar with a pX330 random integration (RI pig). Tissue from muscle, tonsil, spleen, kidney, lymph nodes, oral mucosa, and liver was homogenized (100mg) and RNA was isolated with TRIzol™ Reagent (Invitrogen) according to manufactures' protocol. Two technical replicates were prepared for RT-qPCR (reverse-transcription qPCR). Isolated RNA was digested with 2 U DNAse I for 30 min at 37˚C prior to cDNA synthesis. Synthesis of cDNA was performed according to the protocol with GoScript™ Reverse Transcriptase (Promega). Quantitative PCR was performed with SYBR™ Green master mix (Life Technologies). Primer sequences for Cas9 expression were the following 5' CCCAAGAGGAACAGCGATAAG 3' and 5' CTATTCTGTGCTGGTGGTGG 3'. Differential mRNA expression was calculated by the Relative Standard Curve Method. Cas9 expression was normalised to the reference gene GAPDH (Glyceraldehyde 3-phosphate dehydrogenase). A cDNA dilution from pooled muscle RNA was included on every plate to give standard curves for the calculation of relative expression values for Cas9 and GAPDH.

## Heritability of vector integration

One cloned and transgenic offspring, boar 762–7 (RI pig), was kept for breeding purposes. After reaching sexual maturity, sperm was collected and frozen according to standard practice. In addition, morphology of the sperm was evaluated. A CASA (Computer Assisted Semen Analysis) analysis was run prior to freezing and after thawing. For *in vitro* fertilization (IVF), semen was washed with Androhep® (Minitube) and centrifuged at 600 x g for 6 minutes. To confirm Cas9 integration in semen, sperm was lysed with tail lysis buffer (see above), 0.5% Trition X 100 (Merck), and 40 mM DTT (1,4-Dithiothreitol, Roth) following DNA ethanol precipitation. Cas9 DNA was amplified as described before. Oocyte collection and maturation was performed as describe above. Different sperm concentrations were evaluated for IVF varying from 100–1500 spermatozoa per oocyte. After fertilization, zygotes were cultured in porcine zygote media (PZM-3). Blastocysts were collected on day six, added to 15 μl cell lysis buffer (described previously), and incubated for one hour at 55˚C, to evaluate Cas9 integration. For artificial insemination, semen was diluted 1:1 in Androhep® and transferred twice within 24 hours into a superovulated gilt. The pregnant gilt was sacrificed at day 25 of gestation and fetuses were retrieved from the uterus. DNA was extracted from cephalic parts of the fetuses to detect Cas9 integration.

## *In vitro* activity of Cas9 expression

**Fibroblast isolation.**   Before weaning, fibroblasts were obtained by ear biopsy from transgenic Cas9 positive piglets. Fibroblasts were isolated as previously described [34]. When fibroblasts reached confluency, cells were either frozen or further processed. To obtain Cas9 expressing fetal fibroblasts, boar 762–7 was mated to a wild-type sow, which was slaughtered on day 25 of gestation to retrieve fetuses. Fibroblast cell lines were established from tissues after the removal of excess organs.

**Table 1. Guide RNA sequences to target B2M, GGTA1, and B4GalNT.**

| Gene | Sequence 5'→3' |
|---|---|
| **B2M #2** | GAGTAAACCTGAACCTTCGG |
| **B2M #3** | TGAGTTCACTCCTAACGCTG |
| **B4GALNT2 #3** | ATTGTCTGGGACGTCAGCAA |
| **B4GALNT2 #4** | AGAGTACCACCTCCACAGAG |
| **GGTA1** | CTGACGAGTTCACCTACGAG |

**Guide RNA transfection.**    To evaluate Cas9 protein activity *in vitro*, two cell lines were retrieved from piglets 759–5 and 762–7 (RI pigs) and of piglet 731–1, 732–3, and 733–1 with transposon-based integration. The isolated fibroblasts were transfected with five gRNAs (Table 1). After 762–7 reached sexual maturity and was mated with a wild-type sow, two of the fetal cell lines, 102–12 and 102–14 were also submitted to gRNA transfection. The gRNAs were previously designed and their efficiency tested to induce genomic modifications [35, 36]. The following porcine genes were targeted; beta-2-microglobulin (B2M), Beta-1,4 N-acetylgalactosaminyltransferase 2 (B4GALNT2), and alpha-1,3-galactosyltransferase (GGTA1) to either create an indel (insert and deletion, GGTA1) or a deletion mutation (B2M, B4GALNT2). Guide RNAs were expressed from a plasmid, BPK1520 a gift from Keith Joung (Addgene plasmid # 65777; http://n2t.net/addgene:65777; RRID:Addgene_65777) [37]. Plasmid-based transfection occurred as described earlier.

## Validation of edited cell lines

The edited cells were lysed, and target efficiency and specificity were assessed by flow cytometry and next generation sequencing (NGS).

**Next generation sequencing.**    Knock-out efficiency of gRNA transfected Cas9 expressing fibroblasts was determined by next generation sequencing. The transfected cells were lysed (see above) and B2M, B4GALNT2, and GGTA1 products were amplified by PCR (20 cycles). PCR primers amplifying the target genes are given in S1 Table. Amplicons were purified as described before, and DNA concentrations were determined by the Invitrogen Qubit 4 Fluorometer (ThermoFisher Scientific). DNA of the products was pooled by fragment size to a total concentration of 5 nM and sent for MiSeq sequencing (Illumina). Genome editing efficiency of the generated reads was determined with Geneious Prime Version 2021.0.1. The reads were paired, merged and mapped to the reference gene (NCBI Sus scrofa 11.1).

**Flow cytometry.**    Flow cytometry for B2M and GGTA1 was performed to evaluate the editing efficiency of the integrated Cas9. Flow cytometry to detect expression of B4GALNT2 with *Dolichos biflorus* agglutinin (DBA) in fibroblasts was unsuccessful.

Lectin -based flow cytometry was performed for GGTA1 edited cells to detect α-galactose expression [38]. In total 0.5 x $10^6$ modified and unmodified fibroblast of the same cell line were stained with GSL I-B$_4$ isolectin conjugated with DyLight 649 (Vector laboratories) for 5 minutes at 37˚C. A previously isolated GGTA1 knock-out cell line [36] severed as negative control.

In total 0.5 x $10^6$ B2M modified and unmodified cells were incubated with an anti-swine MHC I monoclonal antibody (Kingfisher-Biotech Inc #WS0550S-100), a PE-Vio labelled IgG2ab secondary anti-mouse antibody (Miltenyi Biotec #130-123-498), and a mouse Ig2b kappa isotype (invitrogen #14-4732-85) as control. In addition, a B2M knock-out fibroblast cell line was stained [36].

**Cas9 inhibitor.** Anti-CRISPR proteins can inhibit DNA cleavage activities of the Cas9 protein [39]. An anti-CRISPR (acr) protein AcrIIA4 vector, a gift from Dominik Niopek (Addgene plasmid # 113037; http://n2t.net/addgene:113037; RRID:Addgene_113037) [40] was co-transfected (10 μl 280 ng/μl) with GGTA1 and B2M gRNA expressing BPK1520 vectors into fetal fibroblasts of 102–12 to prove that edits resulted from transgenic Cas9 expression. Inhibition of Cas9 genome editing was measured by flow cytometry for GGTA1 and B2M as described before.

**Off-targets.** In total 15 potential off-target regions were amplified. For each gRNA three of the most likely off-target sequences were selected with CRISPOR (http://crispor.tefor.net/) and validated by PCR (S3 Table) and Sanger sequencing. Sequences were aligned to reference sequences (NCBI sus scrofa 11.1).

## Results

### Generation of Cas9 expressing pigs

Fibroblasts modified with SB transposon-based Cas9 integration were used for SCNT and transferred into six gilts (Table 2). Out of the six gilts four established pregnancies and delivered 22 SB piglets. Of the 22 piglets 20 were born alive but nine had to be euthanized due to low birth weight and leg deformities related to the SCNT process.

Two gilts were subjected to embryo transfers with fibroblasts modified with random Cas9 integration (Table 2). In total, 15 RI piglets were born. One was born dead and two of the remaining piglets had to be euthanized due to SCNT related health issues (low birth weight and leg deformities). All others were healthy and developed normally.

### Genotyping founder animals

Two SCNT recipients gave birth to 15 piglets (RI pigs) but surprisingly, only two (759–5 and 762–7) carried a Cas9 integration (Fig 2). In contrast, from the transposon integration of which four sows gave birth to 22 piglets (SB pigs) and 15 were positive for Cas9 integration. Purified PCR amplicons were sent for sequencing and aligned to Cas9 sequence (S1 Fig).

**Reverse-transcription qPCR of organ tissue.** Cas9 transcription was confirmed by RT-qPCR in muscle, tonsil, spleen, kidney, lymph nodes, oral mucosa, and liver. Cas9 expression was normalised to GAPDH and fold changes were calculated. Tissue with random integration of Cas9 showed lower Cas9 expression compared to transposon-based integration, except in liver and tonsils (Table 3).

### Heritability of vector integration

Boar 762–7 (RI pigs) was kept for breeding purposes. After IVF, twelve blastocysts were analyzed to investigate transmission of the Cas9 transgene to the next generation. Ten out of 12

**Table 2. Quantitative result from somatic cell nuclear transfer with transposon-based integration (SB pigs).**

| | Transposon-based integration | Random integration |
|---|---|---|
| | **Number of animals** | |
| Transfers | 6 | 2 |
| Pregnancies | 4 | 2 |
| Zygotes/sow (Total number of zygotes transferred) | 59–85 (454) | 80 (160) |
| Piglets (Cloning efficiency %) | 22 (4.7–10%) | 15 (8.8–10%) |

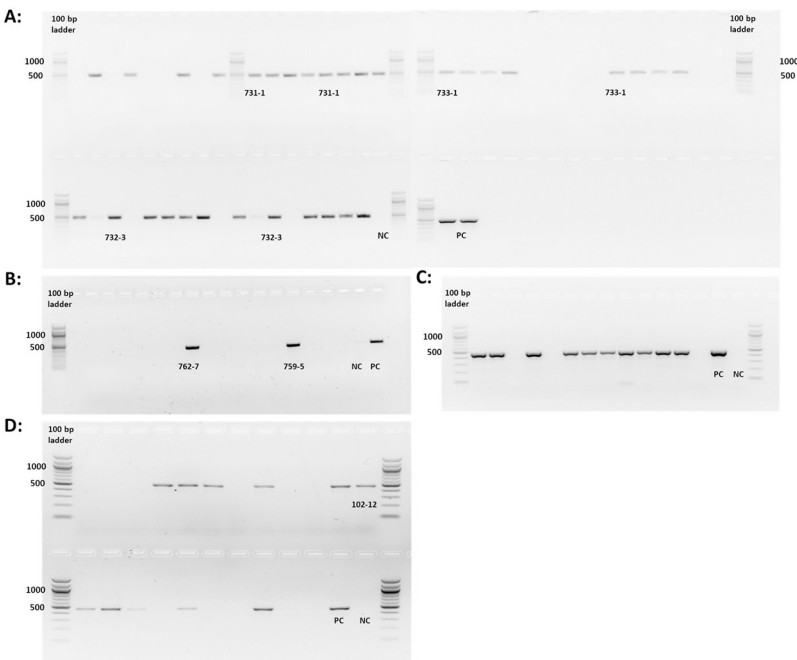

**Fig 2.** Cas9 integration in founder animals, blastocysts and fetuses (Amplicon 500 bp): A: Transposon-based Cas9 integration in founder piglets. B: Random Cas9 integration in founder piglets. C: Cas9 amplification in single blastocysts and D: fetuses) sired by 762–7 (random Cas9 integration).

blastocysts revealed a Cas9 integration (Fig 2). One superovulated gilt was artificially inseminated with semen from boar 762–7, (CASA results and Cas9 integration of semen are shown in S2 Fig and S2 Table. On day 25 of gestation, 21 fetuses were retrieved. Genomic analysis revealed Cas9 integration in 11 fetuses (Fig 2).

**Table 3. Cas9 transcription of isolated organ tissue.**

| Organ | Integration approach | Normalised Cas9 expression[1] | Fold change (RI:SB)* |
|---|---|---|---|
| **Muscle** | Transposon integration | 1.23 | 0.60 |
| | Random integration | 0.74 | |
| **Tonsil** | Transposon integration | 0.36 | 1.18 |
| | Random integration | 0.42 | |
| **Spleen** | Transposon integration | 0.28 | 0.12 |
| | Random integration | 0.03 | |
| **Kidney** | Transposon integration | 1.53 | 0.22 |
| | Random integration | 0.34 | |
| **Lymph nodes** | Transposon integration | 0.33 | 0.22 |
| | Random integration | 0.07 | |
| **Oral mucosa** | Transposon integration | 2.14 | 0.36 |
| | Random integration | 0.78 | |
| **Liver** | Transposon integration | 0.89 | 3.87 |
| | Random integration | 3.43 | |

1 $\frac{relative\ expression\ levels\ value\ Cas9}{Ct\ value\ GAPDH}$

* Fold changes were compared between tissues isolated from the same organs e.g., liver tissue from random integration (RI) against Sleeping Beauty (SB) transposon-based integration.

## *In vitro* activity of Cas9 expression

Isolated fibroblasts of pig 759–5, 762–7, 731–1, 732–3, and 733–1 were transfected with gRNA expressing vectors targeting B2M, B4GalNT, and GGTA1.

**Next generation sequencing.** Pooled samples of PCR amplicons were sequenced with MiSeq and editing efficiencies were calculated. Sequences for B2M were most abundantly represented in the samples (Table 4). NGS data for B2M revealed an editing efficiency of 7.2% and 3.2% for cells isolated from pigs 759–5 and 762–7, respectively. B2M editing efficiency for transposon-based Cas9 integration from isolated cells from pigs 731–1, 732–3, and 733–1 ranged from 2.7–27.6%. Sufficient coverage for GGTA1sequences was obtained for the cells retrieved from pigs 759–5 and 762–7 with editing efficiencies between 2.7 and 2.9%. GGTA1 reads from isolates of 731–1, 732–3, and 733–1 were underrepresented. Sufficient reads were generated for B4GALNT2 in cells isolated from pigs 731–1, 732–3, and 733–1, with editing efficiency of 36.9%, 60.2%, and 51.2%, respectively.

**Flow cytometry GGTA1.** Phenotypic modifications of transgenic Cas9 fibroblasts were assessed after transfection of gRNAs targeting GGTA1 by measuring expression of α-galactose. A GGTA1 knock-out cell line served as negative control. MFI (Median fluorescent intensity) for SB pig cell lines 731–1, 732–3, and 733–1 declined by 5.44, 18.59, and 45.59%, respectively (Table 5 and Fig 3A). As shown in Table 5 and Fig 3B the GGTA1 gRNA transfected cells of RI pigs 759–5 and 762–7 had a decreased MFI of 9.45% and 9.34% compared to untreated cells. Transgenic fetal fibroblasts (102–12 and 102–14) sired by 762–7 (RI pig) were also transfected with a gRNAs targeting GGTA1. Compared to the untreated controls the MFI decreased by 43.87% from cells retrieved from 102–12 and by 56.34% in 102–14 cells (Table 5 and Fig 3C).

**Flow cytometry B2M.** Similar measurements were made for the expression of MHC-I. Cas9 expressing cells transfected with two gRNAs targeting B2M were stained with a swine MHC-I antibody (Fig 4 and Table 6). A B2M knock-out cell line served as negative control Fibroblasts of RI pig 759–5 showed a reduction of 52.44% in MFI and cells from RI pig 762–7

**Table 4. Next generation sequencing editing efficiency and coverage mean.**

| Transfected cells | Integration approach | Target gene | Editing efficiency % | Coverage mean |
|---|---|---|---|---|
| 759–5 | Random integration | B2M | 7.2 | 17322 |
| | | GGTA1 | 2.7 | 1855 |
| | | B4GALNT2 | 30.7 | 499 |
| 762–7 | | B2M | 3.2 | 25469 |
| | | GGTA1 | 2.9 | 3280 |
| | | B4GALNT2 | 19.4 | 719 |
| 731–1 | Transposon-based integration | B2M | 27.6 | 7358 |
| | | GGTA1 | 0.1 | 209 |
| | | B4GALNT2 | 36.9 | 1390 |
| 732–3 | | B2M | 2.7 | 13039 |
| | | GGTA1 | 0.3 | 947 |
| | | B4GALNT2 | 60.2 | 6298 |
| 733–1 | | B2M | 3.4 | 18097 |
| | | GGTA1 | na[*] | na[*] |
| | | B4GALNT2 | 51.2 | 3209 |

[*] No reads were generated

**Table 5. Median fluorescent intensity (MFI) of GGTA1 gRNA transfected cells.**

| Cell line | MFI GGTA1 knock-out* cells | MFI untreated cells[1] | MFI gRNA transfection cells | MFI reduction (%)+ |
|---|---|---|---|---|
| 759–5 | 4.17 | 13.23 | 10.01 | 1.25 (9.45) |
| 762–7 | 4.17 | 9.42 | 8.54 | 0.88 (9.34) |
| 731–1 | 4.17 | 13.42 | 12.69 | 0.73 (5.44) |
| 732–3 | 4.17 | 19.37 | 15.77 | 3.6 (18.59) |
| 733–1 | 4.17 | 20.64 | 11.23 | 9.41 (45.59) |
| 102–12 | 3.98 | 37.18 | 20.87 | 16.31 (43.87) |
| 102–14 | 3.98 | 35.02 | 15.29 | 19.73 (56.34) |

* Negative control

[1] Positive control

+ Reduction between untreated and gRNA transfection

were reduced by 46.81%. MFI of cells from pigs 731–1, 732–3, and 733–1 (SB pigs) decreased by 17.09, 55.79, and 40.85%, respectively. MFI of fetal fibroblast of 102–12 decreased by 47.43% and of 102–14 by 24.30%.

**Flow cytometry Cas9 inhibitor.** Fetal Cas9 expressing fibroblasts were transfected with the anti-CRISPR AcrIIA4 to inhibit transgenic Cas9 activity and to prove that genome edits resulted from the transgenic Cas9 expression. It was expected that in transgenic Cas9 cells transfected with AcrIIA4 and gRNAs, genome editing of the transgenic Cas9 would be inhibited to a certain extent by AcrIIA4. Fetal fibroblasts from 102–12 were transfected with AcrIIA4and gRNAs (AcrIIA4 inhibitor) or gRNA only (gRNA transfection) Untreated cells served as positive control. Inhibition of Cas9 activity was indicated by a higher MFI for cells treated with gRNA for GGTA1 and AcrIIA4 (64.34) compared to gRNA only treated cells (49.66) (Table 7 and Fig 5). Cas9 activity was inhibited by AcrIIA4 in B2M gRNA treated cells with an MFI of 26.20 compared to 16.44 MFI in only gRNA treated cells.

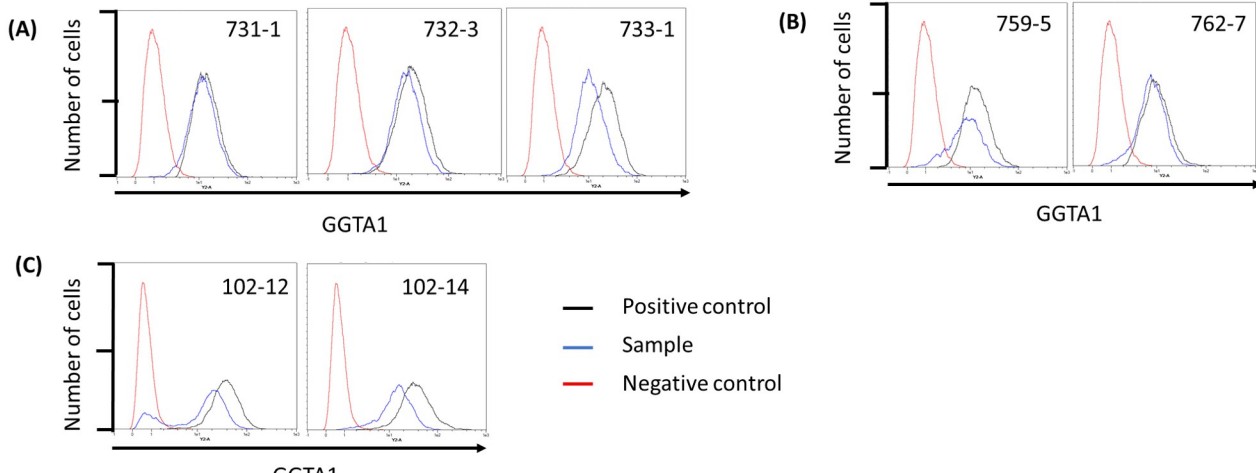

**Fig 3. Flow cytometry of GGTA1 gRNA transfected Cas9 expressing fibroblasts. (A)** Fibroblasts isolated from transposon-based Cas9 integration founder animals 731–1, 732–3, and 733–1 and transfected with GGTA1 gRNA. **(B)** Fibroblasts isolated from random integration founder animals 759–9 and 762–7 and transfected with GGTA1 gRNA **(C)** Isolated Cas9 expressing fetal fibroblasts transfected with GGTA1 gRNA.

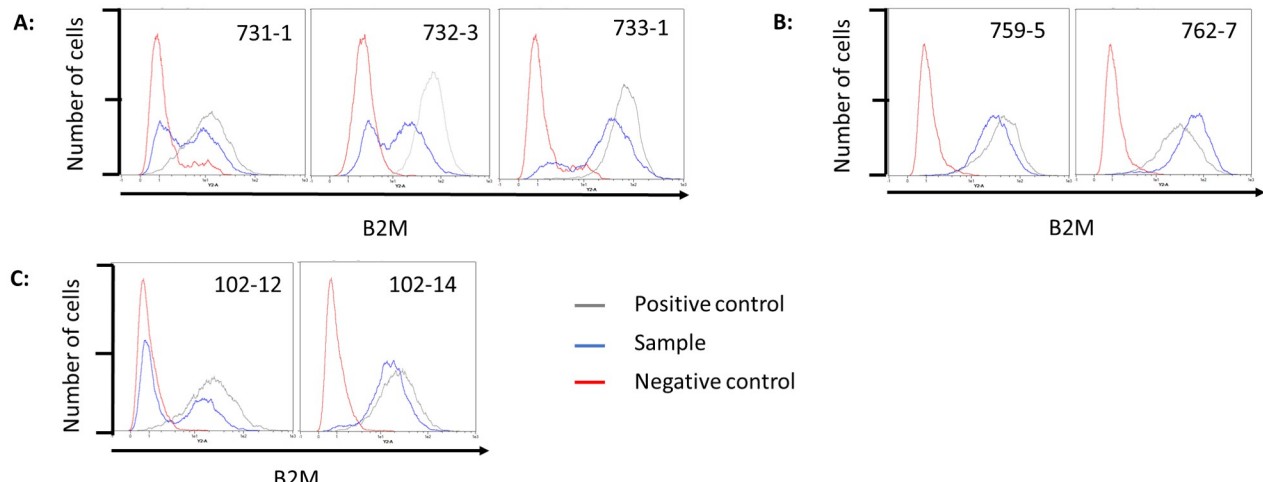

**Fig 4. Flow cytometry of B2M gRNAs transfected Cas9 expressing fibroblasts. (A)** Fibroblasts isolated from transposon-based Cas9 integration founder animals 731–1,732–3, and 733–1. **(B)** Fibroblasts isolated from random integration founder animals 759–9 and 762–7 **(C)** Isolated Cas9 expressing fetal fibroblasts transfected with B2M gRNAs.

**Table 6. Median fluorescent intensity (MFI) of B2M gRNA transfected cells.**

| Cell line | MFI B2M knock-out cells[*] | MFI untreated cells[1] | MFI gRNA transfection cells | MFI reduction (%)[+] |
|---|---|---|---|---|
| **759–5** | 10.74 | 65.66 | 31.23 | 34.43 (52.44) |
| **762–7** | 10.74 | 64.48 | 34.30 | 30.18 (46.81) |
| **731–1** | 11.32 | 15.68 | 13.00 | 2.68 (17.09) |
| **732–3** | 16.23 | 63.90 | 28.25 | 35.65 (55.79) |
| **733–1** | 11.32 | 67.86 | 40.14 | 27.72 (40.85) |
| **102–12** | 16.72 | 32.74 | 22.22 | 10.52 (47.43) |
| **102–14** | 16.72 | 30.87 | 23.23 | 7.5 (24.30) |

[*] Negative control

[1] Positive control

[+] Reduction between untreated and gRNA transfection

**Table 7. Median fluorescent intensity (MFI) after Cas9 inhibition.**

| Cell line | MFI knock-out cells[*] | MFI untreated cells[1] | MFI gRNA transfection cells | MFI reduction (%)[+] | MFI AcrIIA4 inhibitor and gRNA | MFI reduction after inhibition (%)[±] |
|---|---|---|---|---|---|---|
| **102–12 GGTA1** | 21.86 | 82.25 | 49.66 | 32.59 (39.55) | 64.34 | 17.91 (21.78) |
| **102–12 B2M** | 15.74 | 33.59 | 16.44 | 17.15 (51.06) | 26.20 | 7.39 (22.00) |

[*] Negative control (Isolated fibroblasts from GGTA1 and B2M knock-out pig)

[1] Positive control

[+] MFI reduction between untreated cells and gRNA transfection cells

[±] MFI reduction between untreated cells and after inhibition

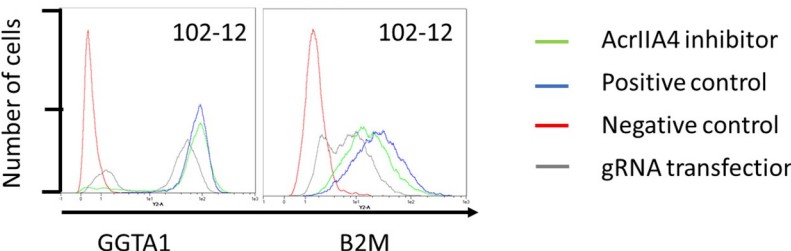

**Fig 5. Flow cytometry of Cas9 inhibitor transfected fibroblasts.** Isolated Cas9 expressing fetal fibroblasts were transfected with expression constructs for AcrIIA4, a Cas9 inhibitor and with gRNA targeting B2M or GGTA1.

## Off-targets

Sanger sequencing and alignment to reference sequence after PCR revealed no off-target mutations in the Cas9 expressing cell lines (S3–S7 Figs).

## Discussion

Genetically modified pigs pose great opportunities for biomedical research. Due to their close resemblance in physiology and anatomy to humans, pigs are more suitable for human disease modeling compared to rodent models. However, the generation of genetically modified pigs to model diseases human-like require great efforts compared to rodent models. Mainly due to the lack of adequate porcine pluripotent stem cells [41], the production of modified pigs relies on SCNT, microinjection or electroporation of zygotes. Somatic cell nuclear transfer remains inefficient and laborious [42] with only 1–3% success rates [43]. By developing Cas9 expressing breeding lines, germline modifications may be avoided, and the pigs can be used to establish diverse disease models. In this study, we generated Cas9 expressing founder animals based on transposon and random integration. Fibroblasts isolated from the transgenic founder animals were subjected to *in vitro* gRNA transfection and one RI boar was bred to a wild-type sow. RNA isolated from muscle, tonsil, spleen, kidney, lymph nodes, oral mucosa, and liver confirmed Cas9 transcription in organs. As in previously generated Cas9 expressing pigs [21, 26], our pigs showed Cas9 genome editing events upon transfection with gRNAs. NGS results revealed editing efficiencies ranging from 0.1 to 2.9% for GGTA1, 3.2 to 27.6% for B2M, and 19.4 to 60.2% for B4GALNT2. It is well established that editing efficiency varies among target loci, as it was also shown in Cre-dependent Cas9 expressing pigs. The study experienced editing efficiencies of 8.1%, 20.2%, and 78.8% for the APC, BRCA1, and BRCA2 loci, respectively, after *in vivo* transfections of gRNAs [21]. Transposon integration is associated with multicopy integration of transgenes [44–46]. Therefore, a multiple Cas9 integration by a SB transposon system was expected to result in higher Cas9 expression. As it was shown in organ tissue, Cas9 expression was overall higher in tissue isolated from the sacrificed SB pig compared to the RI pig. We tried to define copy number and integrations site, to determine copy number differences of transposon-based and random integrations. Nanopore sequencing was performed but with the generated data it was not possible to determine genomic location or copy number of the transgenes.

In addition, we investigated potential off-target mutations which could have been induced by the transfected gRNAs. Due to the constant Cas9 expression in the fibroblasts, there is an increased risk for off-target cleavage. Off-targets could have unwanted effects on the genotype of the pigs which could jeopardize the reliability of the disease model. For each gRNA, the three most potential off-target sites in the genome were selected, none of the 15 targets indicated illegitimate cleavage activity.

Furthermore, transgenic Cas9 expression may led to activation of the adaptive immune system. Studies which investigated Cas protein as therapeutics have observed specific immune responses towards Cas9 in mice [47–49]. However, besides SCNT related health issues our animals grew up healthy. Also, in line with previously generated Cas9 expressing pigs [21, 26] the integration and expression of Cas9 had no negative consequences on fertility [21, 26]. Similar results were obtained from a Cas9 expressing mouse model to study a variety of diseases and biological functions [20]. In addition, random integration of Cas9 in chickens based on transposon integration [50] or phiC31 integrase [26] did not result in any negative side effects.

## Conclusion

In conclusion, we generated functional Cas9 expressing pigs which remained fertile and healthy and therefore are suitable for establishing a Cas9 breeding line. Genome editing of isolated Cas9 expressing fibroblast was feasible, paving the way to generate porcine genotypes for biomedical inquiries. In first investigations on off-target mutations caused by transfected gRNAs were not detected. However, a more detailed investigation of off-target mutations in Cas9 expressing pigs would be necessary when developing human disease models.

## Supporting information

**S1 Fig. Sanger sequence of amplified DNA isolated from 759–5,762–7 (RI pigs), 731–1, 732–3, 733–1 (SB pigs) and aligned to Cas9 reference sequence.** Amplified DNA products can be found in Fig 2(A) and 2(B).
(TIF)

**S2 Fig. Cas9 integration of semen: Cas9 amplification of semen DNA retrieved from boar 762–7, positive control (PC), wild-type DNA, and negative control (NC).**
(TIF)

**S3 Fig. GGTA1 gRNA off-target alignments.**
(TIF)

**S4 Fig. B2M gRNA #2 off-target alignments.**
(TIF)

**S5 Fig. B2M gRNA #3 off-target alignments.**
(TIF)

**S6 Fig. B4GALNT2 #3 off-target alignments.**
(TIF)

**S7 Fig. B4GALNT2 #4 off-target alignments.**
(TIF)

**S1 Table. Primers for detecting genome edits.**
(DOCX)

**S2 Table. Results from computer assisted sperm morphology for boar 762–7.**
(DOCX)

**S3 Table. Primer for off-target regions.**
(DOCX)

## Acknowledgments

We would like to acknowledge the Research Core Unit Genomics (RCUG) at the Hannover Medical School for generating the next generation sequencing data. We are grateful for the competent SCNT and IVF team Andrea Lucas-Hahn, Petra Hassel, Roswitha Becker, and Maren Ziegler. Also, we would like to thank the team of the German Gene bank and the staff of the experimental piggery. Jenny-Helena Söllner was funded within the framework of an intrainstitutional African Swine Fever research consortium.

## Author Contributions

**Conceptualization:** Jenny-Helena Söllner, Hendrik Johannes Sake, Björn Petersen.

**Data curation:** Jenny-Helena Söllner, Antje Frenzel, Rita Lechler, Doris Herrmann.

**Formal analysis:** Jenny-Helena Söllner.

**Funding acquisition:** Björn Petersen.

**Investigation:** Jenny-Helena Söllner, Hendrik Johannes Sake, Antje Frenzel, Rita Lechler, Doris Herrmann.

**Methodology:** Jenny-Helena Söllner, Björn Petersen.

**Project administration:** Björn Petersen.

**Resources:** Walter Fuchs.

**Supervision:** Björn Petersen.

**Validation:** Jenny-Helena Söllner, Hendrik Johannes Sake, Antje Frenzel, Rita Lechler, Doris Herrmann.

**Visualization:** Jenny-Helena Söllner.

**Writing – original draft:** Jenny-Helena Söllner.

**Writing – review & editing:** Hendrik Johannes Sake, Walter Fuchs, Björn Petersen.

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
