## [Decision Letter · Decision Letter 0]

8 Aug 2022

PONE-D-22-03392In vitro genome editing activity of Cas9 in somatic cells after random and transposon-based genomic Cas9 integrationPLOS ONE

Dear Dr. Soellner,

Thank you for submitting your manuscript to PLOS ONE. After careful consideration, we feel that it has merit but does not fully meet PLOS ONE’s publication criteria as it currently stands. Therefore, we invite you to submit a revised version of the manuscript that addresses the points raised during the review process.

We look forward to receiving your revised manuscript.

Kind regards,

Irina Polejaeva, PhD

Academic Editor

PLOS ONE

Journal Requirements:

Reviewers' comments:

Reviewer's Responses to Questions

**Comments to the Author**

1. Is the manuscript technically sound, and do the data support the conclusions?

Reviewer #1: Yes

Reviewer #2: Yes

2. Has the statistical analysis been performed appropriately and rigorously? 

Reviewer #1: N/A

Reviewer #2: Yes

3. Have the authors made all data underlying the findings in their manuscript fully available?

Reviewer #1: Yes

Reviewer #2: Yes

4. Is the manuscript presented in an intelligible fashion and written in standard English?

Reviewer #1: Yes

Reviewer #2: Yes

5. Review Comments to the Author

Reviewer #1: This manuscript focused on the production of transgenic pig cells and animals by using random and transposon-based integration of a constitutive expression of Cas9 protein for in vitro evaluation. They envision that these animals will serve for the production of various animal models of human disease by in vivo therapy with the sgRNA to induce the disease. Their approach aims to overcome the limitations of the SCNT efficiency and unforeseen abnormalities caused by the same procedure. They also expect that Cas9 expressing animals could be further expanded by breeding.

Major: There is no major objection to this manuscript. The manuscript is overall well written and easy to follow.

Minor: Some sentences could be improved such as:

Line 53. Instead of ‘Humans show a higher anatomical…’, use ‘Pigs show a higher anatomical… resemblance to human than rodents, which still…’. When we talk about animal models, we refer to them as references/models for humans. The way how it is written seems a bit the opposite.

Missing reference in Line 82. ‘In mice, several attempts of in vivo tissue modifications… successful (Reference).’

Some typos also need to be addressed, such as Buffer instead of ‘puffer’; ‘Promoter instead of promotor’; ‘SYBR instead of SYBER’.

Also, some sentence clarifications, such as Line 317: ‘GGTA1 reads from isolates of 731…’ to ‘GGTA1 reads isolated from 731…’ or ‘ GGTA1 reads from 731…’

Line 366: ‘Inhibition of Cas9 activity was inhibited which was indicated…’ to ‘Inhibition of Cas9 activity was indicated…’

Technical clarifications: Line 409: ‘results were inconclusive.’ Due to? I think the authors may clarify the reason why these results were inconclusive.

Note that off-targets do not imply off-target mutations. Please, the authors should clarify when they refer to off-target sites and when they refer to off-target mutations.

Improve the conclusion to cover their major accomplishments and clarify the lines below as it was a side experiment to search for most likely, but not all, off-target mutations.

Clarify/modify Line 430 – 432: This off-target conclusion seems a premature answer given that their analysis only searched the three most likely off-target sites and does not present a deep unbiased method of genomic search of off-targets, although I agree that Cas9 expressing animals are suitable for in vivo genome editing.

Reviewer #2: General comments:

In this study, the authors generated Cas9 expression boars by using somatic cell nuclear transfer (SCNT) and random integration/transposon approaches. Cas9 expression was confirmed in different organs of Cas9 transgenic animals. In addition, the authors proved that the integrated Cas9 remained functional. They detected in vitro gene editing, at varied efficiencies of 2-60% using next generation sequencing, in fibroblasts isolated from transgenic founders after transfection with guide RNAs targeting three different genes.

To develop Cas9 expression breeding pigs avoids germline modifications during the generation of animal models using SCNT or embryonic injection methods, especially when KO of certain genes is detrimental to the development of embryos. The strategy reported in this paper enables direct in vivo gene modifications and/or evaluation of in vivo gene therapies. Undeniably, similar Cas9-expressing pigs have been reported in Journals of Genome Res in 2017 and in PNAS, 2021 (Ref. 21 and 26) by other labs. However, the present study enriches data and knowledges in this direction and facilitates further work for in vivo gene editing in animals or humans.

Other comments:

1. L32: ‘such as diabetes’ seems unrelated to the present study, I suggest deleting it.

2. L268 and L274: Table 2 and 3 could be combined to facilitate direct comparison between two transgenic methods, random integration and transposon.

3. L299-301: The sentence, ‘After IVF … to the next generation’, needs to be improved.

4. L316: Should ‘ranged from 3.4 – 27.6 %’ be ‘ranged from 2.7 - 27.6 %?

5. L281: In my view, it is unnecessary to present all sequencing results of 5 founders in Fig 2, since all the sequencing results are the same as that of Cas9. Basically, the authors inclined to report the PCR identification results in this figure. The gel electrophoresis picture could be enough to prove Cas9 integration in F0 animals in Fig 2, and the Cas9 sequencing results could be placed in the supplements. Also, Fig 1, 2, and 3 could be combined to systematically present the PCR identification results for embryos/animals at different generations.

6. L345: Typo. ‘control’ should be ‘control.’

7. L348: Typo. ‘decrease’ should be ‘decreased’

8. L364 – 365: The sentence is confusing to me and could be improved.

9. L407: Is ‘scarified’ correct here?

10. L410: ‘off all gRNAs’ should be ‘of all gRNAs’

11. S2 Table: Typo. ‘S1 Table’ should be ‘S2 Table’

12. Combine the Tables S3, S4, and S5.

6. PLOS authors have the option to publish the peer review history of their article (what does this mean?). If published, this will include your full peer review and any attached files.

Reviewer #1: **Yes: **Iuri Viotti Perisse

Reviewer #2: No

---

## [Author Response · Author response to Decision Letter 0]

31 Oct 2022

Dear Editor and Reviewers,

We gratefully acknowledge your reviewing comments on our manuscript ‘In vitro genome editing activity of Cas9 in somatic cells after random and transposon-based genomic Cas9 integration’ by Soellner et.al. Please find enclosed our revised version of the manuscript.

We thank you for your valuable and helpful comments and remarks regarding our manuscript. We have accepted and acted on your suggestions and provided point-by-point explanations below. We are confident we have addressed the issues raised by you and thereby increased the quality of the revised manuscript to make it acceptable for publication in PLOS ONE.

Here is our point-by-point explanation to your comments and suggestions.

Editors suggestions:

• Comment 1: We note that the grant information you provided in the ‘Funding Information’ and ‘Financial Disclosure’ sections do not match.

Response: Thank you for pointing out the mismatch in the ‘Funding information’ and ‘Funding disclosure’. JHS was funded within the framework of an intrainstitutional African Swine Fever research consortium as stated in the ‘Funding information’.

• Comment 2: We note that you have stated that you will provide repository information for your data at acceptance. Should your manuscript be accepted for publication, we will hold it until you provide the relevant accession numbers or DOIs necessary to access your data. If you wish to make changes to your Data Availability statement, please describe these changes in your cover letter and we will update your Data Availability statement to reflect the information you.

Response: We would like to change the data availability statement to ‘All data is fully available upon request’. Contact for all available data would be bjoern.petersen@fli.de. 

• Comment 3: In your cover letter, please note whether your blot/gel image data are in Supporting Information or posted at a public data repository, provide the repository URL if relevant, and provide specific details as to which raw blot/gel images, if any, are not available.

Response: All raw gel images are available in the ‘Supporting information’ section. 

• Comment 4: Please review your reference list to ensure that it is complete and correct. If you have cited papers that have been retracted, please include the rationale for doing so in the manuscript text or remove these references and replace them with relevant current references. Any changes to the reference list should be mentioned in the rebuttal letter that accompanies your revised manuscript. If you need to cite a retracted article, indicate the article’s retracted status in the References list and include a citation and full reference for the retraction notice.

Response: One additional reference was added to the reference list. Pawluk A, Davidson AR, Maxwell KL. Anti-CRISPR: Discovery, mechanism and function, (39).

Reviewer #1 minor suggestions:

• Comment 1: Some sentences could be improved such as:

Line 53. Instead of ‘Humans show a higher anatomical…’, use ‘Pigs show a higher anatomical… resemblance to human than rodents, which still…’. When we talk about animal models, we refer to them as references/models for humans. The way how it is written seems a bit the opposite.

Response: Thank you for suggesting that some sentences should be rewritten to avoid any confusions. We have rewritten the sentence in line 53 and went through the manuscript again to improve clarity.

• Comment 2: Missing reference in Line 82. ‘In mice, several attempts of in vivo tissue modifications… successful (Reference).

Response: The references for in vivo issue modification in mice were included in the sentence (Reference list 22-24).

• Comment 3: Some typos also need to be addressed, such as Buffer instead of ‘puffer’; ‘Promoter instead of promotor’; ‘SYBR instead of SYBER’.

Response: Thank you for highlighting the typos. We went through the manuscript and corrected the mistakes.

• Comment 4: Also, some sentence clarifications, such as Line 317: ‘GGTA1 reads from isolates of 731…’ to ‘GGTA1 reads isolated from 731…’ or ‘GGTA1 reads from 731…’

Response 4: We agree that the paragraph could have been written clearer. We improved the paragraph and hope it is now clearer to the reader.

• Comment 5: Line 366: ‘Inhibition of Cas9 activity was inhibited which was indicated…’ to ‘Inhibition of Cas9 activity was indicated…’

Response: The line was rewritten.

• Comment 6: Technical clarifications: Line 409: ‘results were inconclusive.’ Due to? I think the authors may clarify the reason why these results were inconclusive.

Response: An explanation why the Nanopore results were inconclusive has been added. ‘Nanopore sequencing was performed but with the generated data it was not possible to determine genomic location or copy number of the transgenes.’

• Comment 7: Note that off-targets do not imply off-target mutations. Please, the authors should clarify when they refer to off-target sites and when they refer to off-target mutations.

o Improve the conclusion to cover their major accomplishments and clarify the lines below as it was a side experiment to search for most likely, but not all, off-target mutations.

o Clarify/modify Line 430 – 432: This off-target conclusion seems a premature answer given that their analysis only searched the three most likely off-target sites and does not present a deep unbiased method of genomic search of off-targets, although I agree that Cas9 expressing animals are suitable for in vivo genome editing.

Response: Thank you for clarifying the impact of the off-target mutations in this paper. We have now improved the terminology and concluded that more detailed investigations are required to rule out any off-target mutations.

Reviewer #2 minor suggestions:

• Comment 1: L32: ‘such as diabetes’ seems unrelated to the present study, I suggest deleting it.

Response: We have deleted the phrase ‘such as diabetes’.

• Comment 2: L268 and L274: Table 2 and 3 could be combined to facilitate direct comparison between two transgenic methods, random integration and transposon.

Response: Thank you for the useful suggestion. We have combined the two tables to one.

• Comment 3: L299-301: The sentence, ‘After IVF … to the next generation’, needs to be improved.

Response: The sentence was rewritten and improved.

• Comment 4: L316: Should ‘ranged from 3.4 – 27.6%’ be ‘ranged from 2.7 - 27.6%?

Response: Thank you for pointing out the mistake, we have corrected it. 

• Comment 5: L281: In my view, it is unnecessary to present all sequencing results of 5 founders in Fig 2, since all the sequencing results are the same as that of Cas9. Basically, the authors inclined to report the PCR identification results in this figure. The gel electrophoresis picture could be enough to prove Cas9 integration in F0 animals in Fig 2, and the Cas9 sequencing results could be placed in the supplements. Also, Fig 1, 2, and 3 could be combined to systematically present the PCR identification results for embryos/animals at different generations.

Response: Thank you for your suggestions. The gel electrophoresis images of the founder animals have now been added and the Sanger sequencing alignments moved to the supplement section. Also, the figures of the gel electrophoresis were combined to one figure.

• Comment 6+7: Typos in L345 and L348

Response: The typos have been corrected

• Comment 8: L364 – 365: The sentence is confusing to me and could be improved.

Response: The sentence was improved and is hopefully clearer for the reader now.

• Comment 9: L407: Is ‘scarified’ correct here?

Response: Thank you for highlighting the word ‘scarified’, we of course meant ‘sacrificed’.

• Comment 10: L410: ‘off all gRNAs’ should be ‘of all gRNAs’

Response: The mistake has been corrected. 

• Comment 11: S2 Table: Typo. ‘S1 Table’ should be ‘S2 Table’

Response: ‘S1 Table’ has been changed to ‘S2 Table’

• Comment 12: Combine the Tables S3, S4, and S5

Response: The tables for the off-target mutation detection have been combined into one table. 

We are looking forward to hearing from you regarding our new submission and to respond to any further comments or questions you may have.

Sincerely yours,

Jenny Söllner

---

## [Editor Report · Decision Letter 1]

1 Dec 2022

In vitro genome editing activity of Cas9 in somatic cells after random and transposon-based genomic Cas9 integration

PONE-D-22-03392R1

Dear Dr. Soellner,

We’re pleased to inform you that your manuscript has been judged scientifically suitable for publication and will be formally accepted for publication once it meets all outstanding technical requirements.

Kind regards,

Irina Polejaeva, PhD

Academic Editor

PLOS ONE
---

## [Editor Report · Acceptance letter]

19 Dec 2022

PONE-D-22-03392R1 

*In vitro* genome editing activity of Cas9 in somatic cells after random and transposon-based genomic Cas9 integration 

Dear Dr. Söllner:

I'm pleased to inform you that your manuscript has been deemed suitable for publication in PLOS ONE. Congratulations! Your manuscript is now with our production department. 

Kind regards, 

on behalf of

Dr Irina Polejaeva 

Academic Editor

PLOS ONE